# Impact of two of the world's largest protected areas on longline fishery catch rates

John Lynham [1]✉, Anton Nikolaev[2], Jennifer Raynor[3], Thaís Vilela[4] & Juan Carlos Villaseñor-Derbez [5]

Two of the largest protected areas on earth are U.S. National Monuments in the Pacific Ocean. Numerous claims have been made about the impacts of these protected areas on the fishing industry, but there has been no ex post empirical evaluation of their effects. We use administrative data documenting individual fishing events to evaluate the economic impact of the expansion of these two monuments on the Hawaii longline fishing fleet. Surprisingly, catch and catch-per-unit-effort are higher since the expansions began. To disentangle the causal effect of the expansions from confounding factors, we use unaffected control fisheries to perform a difference-in-differences analysis. We find that the monument expansions had little, if any, negative impacts on the fishing industry, corroborating ecological models that have predicted minimal impacts from closing large parts of the Pacific Ocean to fishing.

---

[1] Department of Economics, University of Hawaiʻi at Mānoa, Saunders Hall 532, 2424 Maile Way, Honolulu, HI 96822, USA. [2] Information and Computer Sciences, University of Hawaiʻi at Mānoa, 103 Keller Hall, Honolulu, HI 96822, USA. [3] Department of Economics, Wesleyan University, Public Affairs Center 204, 238 Church Street, Middletown, CT 06459, USA. [4] Conservation Strategy Fund, 1636 R St. NW, Suite 3, Washington, DC 20009, USA. [5] Bren School of Environmental Science and Management, University of California, 2400 Bren Hall, Santa Barbara, CA 93106, USA. ✉email: lynham@hawaii.edu

Marine protected areas (MPAs) provide many benefits, such as increasing biomass of commercially important species, protecting biodiversity, sequestering carbon in undisturbed sea bottoms, and mitigating the effects of climate change[1–3]. The potential benefits of MPAs for resident species are obvious (for example, safeguarding habitat and reducing fishing mortality), but MPAs may also benefit highly migratory species by protecting important aggregation or spawning areas[4,5]. MPAs might also impose short-run opportunity costs by displacing existing activities, notably fishing effort, or by preventing the development of new uses, such as deep-sea mining[6,7]. The benefits derived from MPAs are often dispersed across fishers and a diversity of other stakeholders, and take time to develop. Negative impacts tend to be more concentrated on subsets of stakeholders, such as fishers, and may be more immediate[6]. In addition, industry-based cost estimates are often difficult to validate because business information is generally confidential. As a result, the concerns of highly organized and vocal opponents may become more salient in the decision-making process. This asymmetry in information and influence may, in turn, lead to an under allocation of protected areas by eroding existing protections or preventing new conservation interventions.

The Papahānaumokuākea Marine National Monument (PMNM) and the Pacific Remote Islands Marine National Monument (PRI) — the third- and fifth-largest protected areas in the world, respectively — offer a unique opportunity to rigorously examine the economic effects of large MPAs on the fishing industry. PMNM comprises an area of 1,508,870 km$^2$ around the northwestern Hawaiian islands, while PRI comprises 1,277,860 km$^2$ across 5 areas to the south and east of the Hawaiian Islands (Fig. 1a; Supplementary Table 1). U.S. President George W. Bush initially established both monuments, and President Barack Obama more than quadrupled the size of these areas in 2014 (PRI) and 2016 (PMNM). Today, the monuments protect corals, fish, birds, sharks, whales, and other marine life within a combined area larger than the land mass of Argentina. Part of the PRI monument also protects some known spawning grounds for bigeye tuna (*Thunnus obesus*), the most commercially important species in the region[8–10]. The stated purpose of both expansions was to protect flora and fauna within the boundaries of the monuments and not specifically to benefit fisheries operating outside the monuments.

Regulators anticipated that the establishment of PMNM and PRI could potentially lead to economic costs for the Hawaii-based longline fishing industry, by far the most lucrative fishery in the region. Prior to the expansion of PMNM, the fishing industry argued that the expansion would have "significant economic impacts to Hawaii longline participants and seafood consumers" and that potential losses would amount to "$10 million annually in wholesale landed value from Hawaii longline fishery, translating in approximately $30 million across Hawaii's retail seafood market"[11]. This claim appears to be based on the argument that historically about $10 million worth of fish were caught within PMNM each year. Likewise, an internal report by the National Oceanic and Atmospheric Administration (NOAA) predicted that the upper bound on potential annual losses could be $7.8 million "under the assumption that catch from the Northwestern Hawaiian Islands (NWHI) is completely 'lost', which is likely an overly restrictive assumption."[12] The extent to which fishers could adapt, by fishing elsewhere for example, and avoid these costs was uncertain. A major concern is that the monuments exclude commercial fishing in nearly one-third of the country's Exclusive Economic Zone in the Pacific Islands region. Loss of access to this area may increase competition with international fleets, which are allowed to fish on the high seas but not within the U.S. EEZ.

We provide an ex post examination of the impacts of the Obama-era expansions of the PMNM and PRI monuments on U.S.-based fishing operations. In our analysis, we make use of three different data sources. First, we analyze trends in industry-wide catch and revenue from NOAA and fisheries management council logbook summary reports. We find that both catch and revenue have increased since the expansions, but these trends may be due to favorable environmental conditions occurring at the same time. Second, to disentangle the causal effect of the expansions, we analyze trends in catch rates (catch-per-1000-hooks, catch-per-set, catch-per-trip, and catch-per-kilometer-traveled) in fisheries affected by the expansion versus unaffected fisheries, using difference-in-differences regressions. This technique, described further below, uses set-level data from the NOAA Observer Program to construct a control for what would have happened in the affected fisheries in the absence of each expansion. The control fisheries are not a strict control as in a manipulative experiment with random assignment. Instead, they are fisheries that are influenced by the same factors (observed and unobserved) that might be correlated with the creation of the protected areas, such as changes in oceanographic conditions or management rules, but these control fisheries remain unaffected by the new protected areas. We find that the expansions did not have statistically significant negative effects on catch rates, with the exception of a significant drop in catch-per-fishing-trip; the latter is caused by a smaller increase in trip length and total hooks deployed per trip in the affected fishery versus in the control fishery. Finally, we use vessel tracking data from Global Fishing Watch (GFW) to refine and verify our estimates of distance traveled[13]. Overall, we show that two of the largest protected areas on earth had little, if any, economic impact on the U.S.-based fishing industry. These results corroborate ecological models that predict little economic harm from closing large parts of the Pacific Ocean to fishing[14–17].

## Results

**Aggregate trends**. To motivate the likely magnitude of impacts of the monument expansions, we examine how much longline fishing activity was displaced from the expansion area. We focus on the fleet that primarily targets bigeye tuna because these trips account for over 95% of the revenue generated from longline fishing in Hawaii, the main industry predicted to be harmed. We base our analysis on data from the NOAA Observer Program, which collects information on fishing trips, including the location and time of every fishing event (or set) and the number of fish caught for both target and non-target species. The Hawaii-based longline fishery has been monitored under a mandatory federal observer program since February 1994. Trips targeting tunas have approximately 20% random observer coverage, whereas trips targeting swordfish have 100% coverage. We restrict the sample to 2010–2017 in order to exclude the possible impacts of the Great Recession, but all of the results that follow hold if we use the entire observer dataset. Using location information, we classify each fishing set into one of three categories: inside PRI, inside PMNM, or outside both monuments. Data are aggregated to maintain data confidentiality, per NOAA requirements. Figure 1b shows the annual spatial allocation of fishing sets for these trips. At least 90% of fishing sets have historically taken place outside monument waters; the annual percentage of sets inside the monuments ranged from 4 to 9% in the pre-expansion years. The results are similar when looking at catch within the monument waters, which constitutes no more than 10% of fleet-wide annual catch during the study period. We also see no evidence of a blue paradox, defined as a ramp up in fishing effort inside a proposed protected area prior to prohibitions going in to effect[18–20]. This

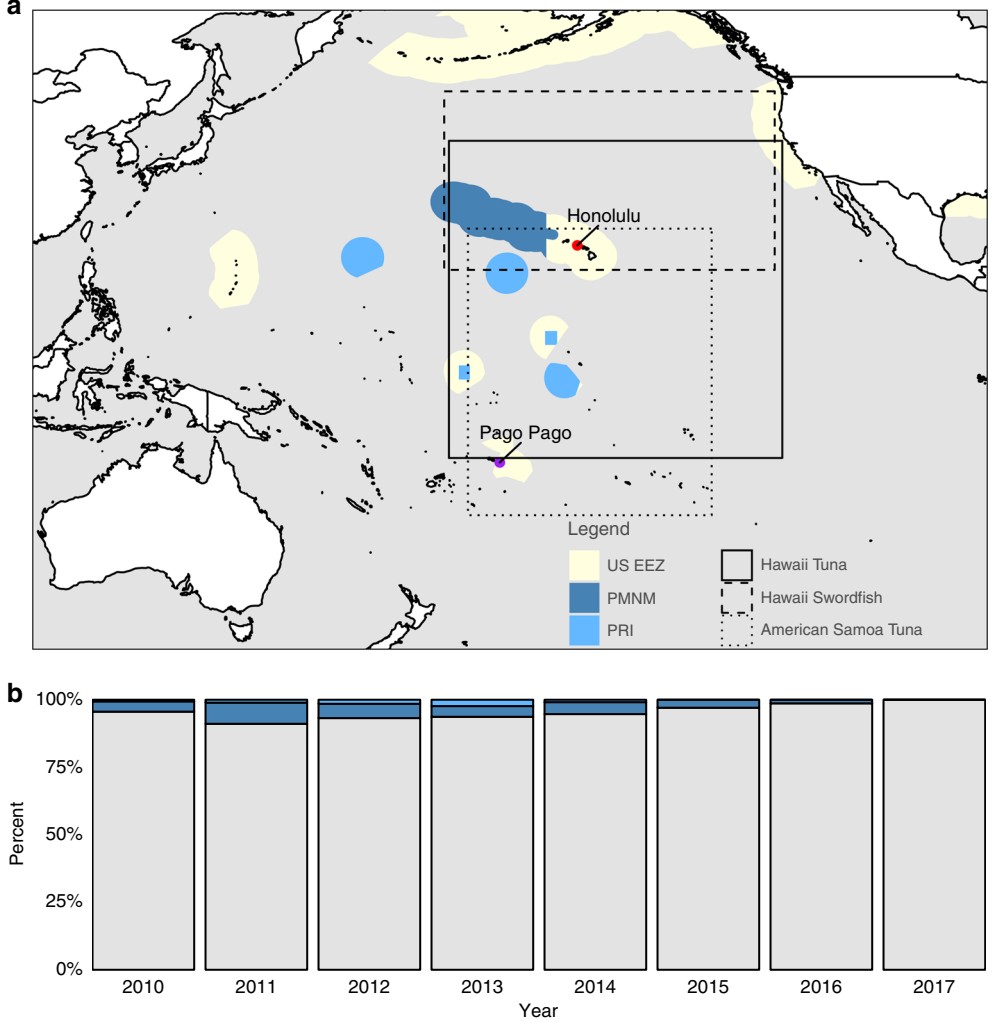

**Fig. 1 Map of the Marine National Monuments and fishing locations by year.** In panel **a**, dark blue indicates Papahānaumokuākea Marine National Monument (PMNM), light blue indicates the Pacific Remote Islands Marine National Monument (PRI), and light yellow indicates the U.S. exclusive economic zone (EEZ). The three rectangles represent the maximum spatial extent of the three fishing fleets analyzed, based on observer data: solid black for the Hawaii Tuna fleet, long dashes for the Hawaii Swordfish fleet, and short dashes for the American Samoa Tuna fleet. The northern boundaries of the rectangles are based on the furthest north set, the eastern boundaries are based on the furthest east set, etc. Thus, the vertices of the rectangles do not necessarily indicate the location of a fishing set. The locations of the two main fishing ports for the fleets analyzed are also displayed. Panel **b** shows the percentage of fishing sets by year for the Hawaii-based tuna fleet that start inside the Pacific Remote Islands Marine National Monument (light blue), inside the Papahānaumokuākea Marine National Monument (dark blue), or outside the monuments (gray).

would suggest that waters outside of the monuments are at least as productive as those inside. Taken as a whole, Fig. 1 suggests that a very small fraction of total fishing effort has been displaced by the monument expansions.

Aggregate data point to a strengthening of the industry following the monument expansions. Based on logbook summary reports, which capture 100% of activity in the longline fleet, revenue, catch, and catch-per-unit-effort (CPUE) have all increased since the expansions (see Figs. 2 and 3; Supplementary Fig. 1). The increases in catch are generally statistically significant, especially following the PRI expansion (Supplementary Table 2). Although the fishery has been regulated with a total allowable catch (TAC) limit on bigeye tuna since 2009, the fleet has been allowed to exceed this by transferring unused TAC allocation from other U.S. territories. These allocations have allowed total catch to increase, despite industry-wide catch limits[21,22]. Aggregate increases in total catch and total revenue may be masking negative impacts from the monuments, such as forcing vessels to fish in less productive areas, to travel further, or to

compete with more vessels in less space, all of which would increase the cost of fishing. Although we do not have data on profits in the industry because we do not have access to either individual or aggregated cost data, we can examine changes in CPUE. CPUE, measured as the ratio of catch to effort, is a proxy for fishery profitability because it relates the costs of fishing to the benefits of fishing. As an example, if the new protected areas cause a vessel to expend twice as much effort to catch the same amount of fish as before, then CPUE would decline by 50%. On the contrary, we observe increases in three aggregate measures of CPUE following each expansion (Fig. 3), using total catch per total hooks, total catch per total fishing sets, and total catch per total fishing trips as the relevant metrics. These increases in CPUE suggest that either less effort yields higher catch or that the same effort now yields more catch.

Using individual vessel data from the observer program and focusing only on the Hawaii-based tuna fleet, we find that both expansions are correlated with an increase in CPUE for target species, measured with the aforementioned effort metrics as well

as catch-per-kilometer-traveled (Fig. 4). We also use a series of regressions with additional controls to document whether these increases are, in fact, statistically significant and not explained by other factors such as changes in environmental conditions. In particular, we account for the fact that oceanographic conditions may have increased catch rates since the expansions began by including contemporaneous and lagged El Niño indexes as control variables. The specific El Niño index we use is the monthly Niño 3.4 index provided by the National Center for Atmospheric Research[23]. Tuna movement and distribution is known to be influenced by El Niño-Southern Oscillation (ENSO) events as fish track the most suitable water temperature[24,25]. The results for catch-per-1000-hooks are shown in Table 1 and for the remaining outcomes in Supplementary Tables 3–5. We find that the statistically significant increases in CPUE sometimes become

non-significant or even negative as additional controls are added to the regression model, highlighting the importance of controlling for confounding factors that may have occurred at the same time as the expansions.

**Causal inference.** We now turn to our most detailed analysis of the possible economic impacts of the expansions on the fishing fleet. A criticism of the literature on marine reserve impacts is that the absence of an appropriate control makes it challenging to disentangle reserve impacts from unobserved factors that are changing at the same time[26]. There are two key assumptions that, if satisfied, will allow us to make causal claims about the effects of the expansions: the excludability and no interference assumptions[27]. The excludability assumption states that any confounding factors or rival explanations have been excluded or controlled for[27]. For example, one well-known MPA study observed a 90% increase in CPUE after MPA implementation, but this was partially due to favorable changes in environmental conditions and a statistically non-significant decline in fishing effort[28,29]. The no interference assumption states that when using a control-impact approach, there can be no interference or spillover from the impacted fishery to the control fishery. This requires that the treated group of vessels does not change the behavior of the control group of vessels following reserve implementation. A classic example of a violation of the no interference assumption would be if vessels impacted by the reserve displace the control vessels and force them to fish in less productive waters, causing their CPUE to decline. This could make the negative impacts of the reserve appear smaller than they actually are. Even the announcement of a future protected area may lead to changes in behavior that complicate evaluation of the protected area once it is established[18]. Similar issues arise in the evaluation of terrestrial protected areas and other fisheries regulations[30–32]. A meta-analysis[27] of nearly 200 studies attempting to make causal claims about the impacts of marine reserves shows that only a few addressed the issues of excludability and no interference, and only one study[33] addressed both.

In order to make credible statements about the causal impacts of the expansions, we need to find a control for CPUE in the Hawaii tuna fishery. In other words, what would the trend in CPUE have been if the monuments had not been expanded? The control must be influenced by the same unobserved factors that might be correlated with the monument expansions, such as changes in oceanographic conditions or management rules, but remain unaffected by the expansions themselves (allowing us to control for these unobserved factors and satisfy the excludability assumption). We capitalize on incidental catch of bigeye and yellowfin tuna (*Thunnus albacares*) in two closely related fisheries

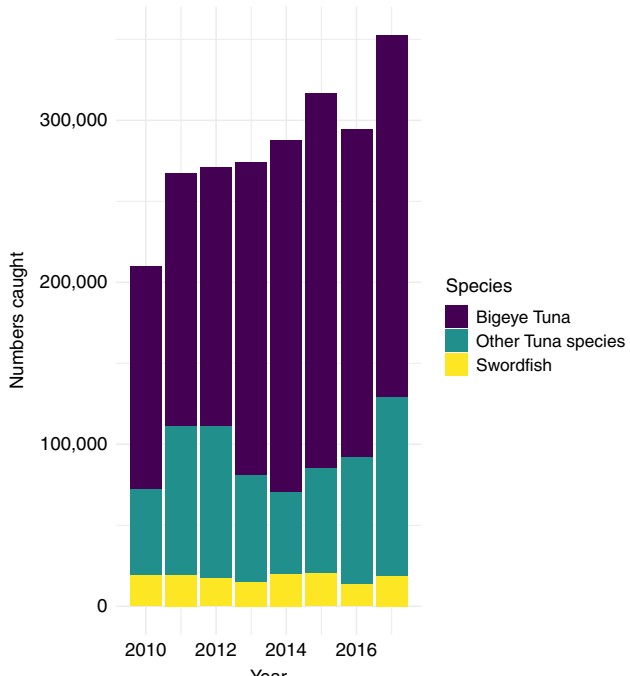

**Fig. 2 Total catch by year for commercial pelagic fisheries.** Source: Annual hawaii limited access longline logbook summary reports from 2010 to 2017. Data is from All Sets (Tuna and Swordfish) and All Areas (inside monuments, inside and outside U.S. EEZ, etc.). The total catch is color-coded by species: Bigeye Tuna in dark purple, Other Tuna Species in dark teal, and Swordfish in yellow. Source data are provided as a Source Data file.

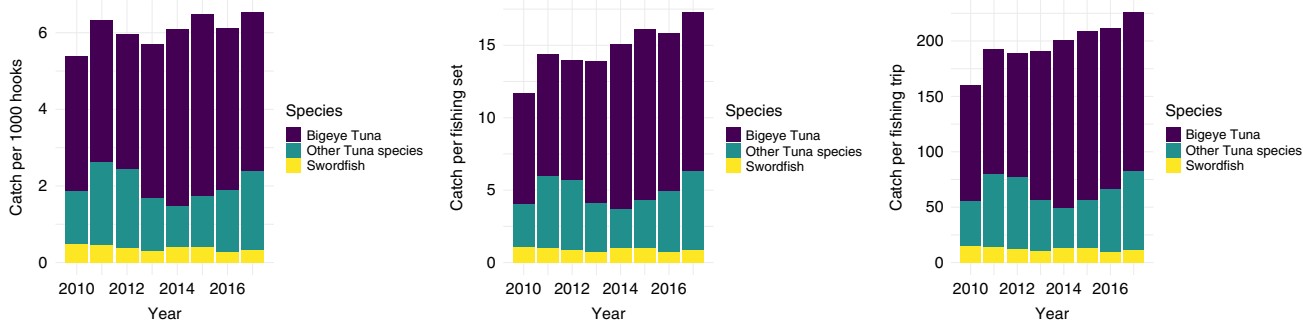

**Fig. 3 Total catch-per-unit-effort by year for commercial pelagic fisheries.** Source: Annual hawaii limited access longline logbook summary reports from 2010 to 2017. Data are from All Sets (Tuna and Swordfish) and All Areas (inside monuments, inside and outside U.S. EEZ, etc.). The total catch is color-coded by species: Bigeye Tuna in dark purple, Other Tuna Species in dark teal, and Swordfish in yellow. Source data are provided as a Source Data file.

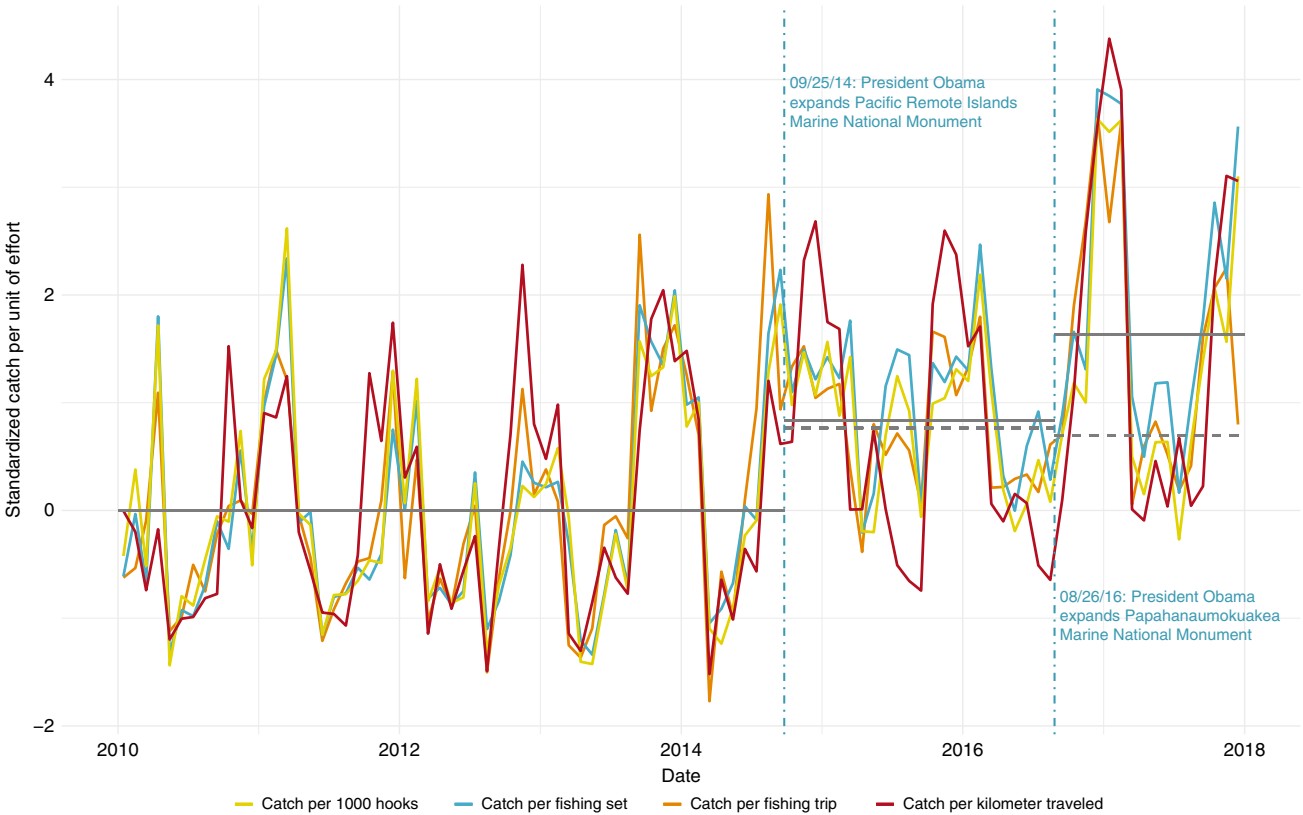

**Fig. 4 Monthly CPUE pre- and post-expansions for the Hawaii-based tuna fleet.** The four catch-per-unit-effort (CPUE) measures are catch-per-1000-hooks (solid yellow line), catch-per-fishing set (solid blue line), catch-per-fishing-trip (solid orange line), and catch-per-kilometer-traveled (solid dark red line). Vertical dashed blue lines indicate the dates of the two expansions. Each time series has been standardized by subtracting its mean value prior to the PRI expansion and dividing by its standard deviation prior to the PRI expansion. The horizontal solid gray lines indicate the mean of all four catch-per-unit-effort measures (i) prior to the expansion, (ii) following the PRI but prior to the PMNM expansion, and (iii) following the PMNM expansion. The horizontal dashed gray lines represent the same calculation (the mean of all four standardized catch-per-unit-effort measures) for two control fisheries: the segment following PRI expansion but prior to PMNM expansion is calculated using Hawaii swordfish fleet data and the post-PMNM segment is calculated using American Samoa tuna fleet data. Both segments are standardized relative to their means and standard deviations prior to the PRI expansion. All lines are drawn for the purposes of motivation and data visualization only. For appropriate tests of statistically significant differences, please refer to the regression tables.

**Table 1 Catch of Bigeye and Yellowfin Tuna per 1000 Hooks.**

|  | (1) | (2) | (3) | (4) | (5) |
|---|---|---|---|---|---|
| Constant | 4.927[***] | 5.732[***] | 5.329[***] | 5.060[***] | −0.355 |
|  | (0.039) | (0.084) | (0.113) | (0.341) | (4.496) |
| PRI expansion | 1.083[***] | 1.049[***] | 0.078 | −0.082 | −0.605[***] |
|  | (0.069) | (0.068) | (0.213) | (0.215) | (0.225) |
| PMNM expansion | 0.887[***] | 0.783[***] | 0.494[***] | 0.494[***] | 0.103 |
|  | (0.095) | (0.094) | (0.186) | (0.188) | (0.254) |
| Month dummies | No | Yes | Yes | Yes | Yes |
| Year dummies | No | No | Yes | Yes | Yes |
| Vessel dummies | No | No | No | Yes | Yes |
| Additional controls | No | No | No | No | Yes |
| Observations | 29,750 | 29,750 | 29,750 | 29,750 | 29,750 |
| $R^2$ | 0.022 | 0.047 | 0.051 | 0.097 | 0.102 |

Notes: Each successive column adds additional controls to a simple regression test of whether catch-per-unit-effort increases following the first expansion and again following the second expansion (Column (1)). The sample runs from January 1, 2010 to December 31, 2017. Heteroskedasticity-robust standard errors presented in parentheses. The additional controls are whether the set included an experimental component, a dummy variable for whether Western and Central Pacific Fisheries Commission waters were closed to fishing, a dummy variable for whether Inter-American Tropical Tuna Commission waters were closed to vessels longer than 24 m, Monthly El Niño indicator, Monthly El Niño indicator lagged by 1 year, Monthly El Niño indicator lagged by 2 years, and Monthly El Niño indicator lagged by 3 years. $^*p < 0.1$; $^{**}p < 0.05$; $^{***}p < 0.01$ for two-sided $t$ test of statistical significance using heteroskedasticity-robust standard errors.

to construct our controls—the Hawaii longline swordfish (*Xiphias gladius*) fleet for the PRI expansion and the American Samoa longline albacore tuna (*Thunnus alalunga*) fleet for the PMNM expansion (the dashed gray lines in Fig. 4 show the mean of the four CPUE measures for these control fisheries).

The Hawaii swordfish fleet is a good control for the PRI expansion for two reasons. First, the swordfish fleet incidentally catches bigeye and yellowfin tuna when targeting swordfish. The current scientific consensus is that bigeye tuna is one large population spread across the Pacific Ocean[34–39] and yellowfin

tuna is three or more large populations[35,40]. Thus, any environmental variation that influences abundance of these tuna species in the Pacific should be reflected in bycatch rates for the swordfish fleet, allowing us to exclude environmental variation as a source of bias. The fact that the swordfish and tuna fleets both fish in the same general region strengthens this argument (Fig. 1a). Supplementary Fig. 4 shows the high degree of correlation in catch rates across the two fisheries. Furthermore, the swordfish fleet is U.S. flagged, subject to many of the same regulations as the Hawaii tuna fleet, and sells their catch at the same auction. This allows us to control for unobserved changes in regulatory, institutional, and market conditions as potential sources of bias in our regression estimates. Second, since 1994 a swordfish set has never been recorded inside PRI by NOAA observers (and the swordfish fleet has 100% observer coverage). Thus, the Hawaii swordfish fleet should not be directly affected by the PRI expansion. CPUE of bigeye and yellowfin tuna in the swordfish fleet should be influenced by environmental, economic, and social factors that also affect the Hawaii tuna fleet but we expect no direct impact from the PRI expansion, allowing us to satisfy the excludability assumption.

When we analyze the PMNM expansion, we use incidental catch of bigeye and yellowfin tuna in the American Samoa albacore fishery as our control measure of CPUE. The American Samoa fishery is also U.S.-flagged and subject to similar regulations as the Hawaii tuna fleet. Again, this fleet is primarily targeting a different species so their incidental catch of bigeye and yellowfin tuna serves as an ad hoc sampling of their abundance in the Pacific. An American Samoa longline permit grants the right to fish around American Samoa, Guam, the Northern Mariana Islands, and the PRI areas, but it has never included the legal right to fish in PMNM (Fig. 1a shows the spatial extent of this fleet). Thus, the American Samoa tuna fleet should be exposed to similar changes in oceanographic conditions as the Hawaii tuna fishery but be unaffected by the expansion of PMNM because the PMNM expansion merely closed waters that have always been closed to the American Samoa fleet (helping us to satisfy the excludability assumption). In Supplementary Fig. 5, we show that the catch rates of bigeye and yellowfin tuna are strongly positively correlated across the Hawaii tuna and American Samoa albacore tuna fisheries, confirming our claim that the American Samoa fleet is a plausible control. It can also be seen in Fig. 2 and in the Supplementary Information file that the timing of both expansions appears to coincide with a general increase in bigeye and yellowfin tuna CPUE; this has been linked to favorable recruitment conditions in 2012. In summary, the excludability assumption is most likely to hold for the Hawaii tuna vs. Hawaii swordfish analysis since these two fleets fish in roughly the same part of the Pacific Ocean. It is least likely to hold for the Hawaii tuna vs. American Samoa albacore tuna analysis because these fleets fish so far apart and environmental conditions are less correlated with increasing distance[41].

Both controls also likely satisfy the no interference assumption. The most obvious mechanism through which interference could take place is if the expansions moved treated vessels (Hawaii tuna trips) into the fishing areas of untreated vessels (Hawaii swordfish and American Samoa albacore tuna trips), thereby influencing their productivity through congestion, information sharing, or localized depletion. We test for whether there is any evidence that interference is taking place using a series of simple tests. We calculate the average distance between the treated and untreated fleets by month and then test whether this decreases following either expansion. We do not find any evidence of decreases in average distance and, in some cases, observe statistically significant increases in distance apart following the expansions (Supplementary Table 7). For PRI, the average distance between

the treated and untreated fleets prior to either expansion exceeds 1700 km and for PMNM it is larger than 4000 km (which is roughly the distance between Hawaii and American Samoa). These robustness checks give us confidence that the no interference assumption holds, and that each of the surrogate fisheries are valid controls. In contrast to the excludability assumption, the no interference assumption is most likely to hold for the Hawaii tuna vs. American Samoa albacore tuna analysis since these fleets operate out of separate ports and rarely, if ever, directly interact with each other. The no interference assumption is least likely to hold for the Hawaii tuna vs. Hawaii swordfish analysis because these fleets fish out of the same home port, are free to switch from targeting tuna to targeting swordfish, and their fishing grounds overlap more.

We combine the Hawaii tuna and Hawaii swordfish observer data to evaluate the PRI expansion and then combine the Hawaii tuna and American Samoa albacore tuna observer data to evaluate the PMNM expansion. Thus, we use two separate datasets to evaluate each expansion separately. We perform two sets of difference-in-differences regressions, akin to a before-after-control-impact (i.e., BACI) design in experimental ecology[42]. The main regression equation we estimate is explained in the Methods section. The key co-variate of interest is an indicator variable for the expansion interacted with an indicator variable for the Hawaii tuna fleet. This term shows the impact of the expansion, controlling for any changes in CPUE that would have occurred if the expansion had never taken place (the counterfactual trend).

Even after controlling for the counterfactual trend in bigeye and yellowfin CPUE, we observe an increase in CPUE following either expansion (Table 2). In all but one regression model, this increase is statistically significant. In Supplementary Tables 8–10, we replicate these results for the three other measures of CPUE and observe statistically significant increases in 12 models and statistically non-significant changes in 5 models. The only significant decrease we observe is in one model with catch-per-fishing-trip as the outcome variable. Further investigation reveals that this decrease is due to the length of fishing trips growing more slowly in the Hawaii tuna fleet, relative to the control, following the expansion (Supplementary Table 11).

**Distance robustness check**. One potential drawback of our measure of trip distance using observer data is that we approximate the total distance traveled (Supplementary Figs. 2 and 3; Supplementary Tables 6 and 12). In particular, the actual distance traveled may be higher following the expansions if vessels are forced to do more searching between fishing sets, but this distance traveled would be missing from our imputed measure since it takes place between sets (see Methods section). In order to test whether our results are robust to using more accurate distance information, we make use of detailed fishing vessel locations provided by GFW. GFW uses Automatic Identification System (AIS) messages to track the location and activity of large fishing vessels, in some cases as often as every fifteen seconds. We perform regression-based analysis of daily distance traveled using the GFW data and fail to reject the null hypothesis that distance traveled is unchanged following the PMNM expansion (Supplementary Fig. 7; Supplementary Table 13). We also show that the bias in our approximation of travel distance does not increase following the expansion (Supplementary Table 14). In summary, the GFW data suggests that the Hawaii tuna and swordfish fleets are not being forced to travel further following the PMNM expansion.

## Discussion
By combining three different data sources and accounting for changes in control fisheries, we provide a detailed account of how

**Table 2 Difference-in-differences estimation.**

|  | (1) | (2) | (3) | (4) | (5) | (6) |
|---|---|---|---|---|---|---|
| PRI expansion | 0.338*** (0.044) | −0.357** (0.162) | −0.591*** (0.172) |  |  |  |
| PMNM expansion |  |  |  | 0.945*** (0.287) | −0.195 (0.318) | −0.489 (0.398) |
| Hawaii-based Tuna Trips | 4.037*** (0.044) | 4.169*** (0.046) | 4.218*** (0.093) | 1.236*** (0.083) | 1.091*** (0.086) | 1.980*** (0.338) |
| PRI * Hawaii | 0.745*** (0.082) | 0.782*** (0.082) | 0.564*** (0.098) |  |  |  |
| PMNM * Hawaii |  |  |  | 0.703** (0.298) | 0.754** (0.303) | 0.532 (0.329) |
| Month dummies | No | Yes | Yes | No | Yes | Yes |
| Year dummies | No | Yes | Yes | No | Yes | Yes |
| Vessel dummies | No | No | Yes | No | No | Yes |
| Additional controls | No | No | Yes | No | No | Yes |
| Observations | 33,444 | 33,444 | 33,444 | 34,964 | 34,964 | 34,964 |
| $R^2$ | 0.160 | 0.177 | 0.222 | 0.022 | 0.043 | 0.092 |

Notes: The dependent variable in all columns is Catch of Bigeye and Yellowfin Tuna per 1000 Hooks. In Columns (1)–(3), the control group is Hawaii-based swordfish trips and the sample runs from January 1st 2010 to August 25th 2016. In Columns (4)–(6), the control group is American Samoa-based tuna trips and the sample runs from January 1, 2010 to December 31, 2017. Heteroskedasticity-robust standard errors presented in parentheses. The additional controls are whether the set included an experimental component, a dummy variable for whether the WCPFC waters were closed to fishing, a dummy variable for whether IATTC waters were closed to vessels longer than 24 m, Monthly El Niño indicator, Monthly El Niño indicator lagged by 1 year, Monthly El Niño indicator lagged by 2 years, and Monthly El Niño indicator lagged by 3 years. $*p < 0.1$; $**p < 0.05$; $***p < 0.01$ for two-sided $t$ test of statistical significance using heteroskedasticity-robust standard errors.

the creation of two of the world's largest MPAs has affected the Hawaii-based longline fishing industry to date. The evidence overwhelmingly suggests that the industry experienced little, if any, negative economic impact on catch rates from the expansion of the national monuments. We find no observable declines in catch and, in fact, both aggregate CPUE and revenue increase following each expansion (although the increases are smaller and, in some cases, statistically indistinguishable from zero when we include our control fisheries). These results contradict earlier predictions of large economic losses and show that the two MPAs, though developed with the objective to protect rare iconic species, did not diminish the CPUE in the Hawaii longline tuna fishery. This finding is consistent with previous evaluations of small and medium sized reserves[33,43–45].

Why are we not observing detrimental impacts on the fishing fleet? The first obvious reason is that over 90% of fishing (by number of sets and by total catch) took place outside the monuments prior to the expansion (Fig. 1b). In addition, the Hawaii-based fleet has ample access to unprotected areas on the high seas. Given the high mobility of the main target species, fishing fleets may still harvest these populations in areas open to fishing. In addition, it appears that alternative fishing grounds are at least as productive as those inside the monuments, as corroborated by both our CPUE analysis (Tables 1 and 2; Supplementary Tables 3–5 and 8–10) and the fact that we do not observe a preemptive rush to fish in the monuments prior to their anticipated closure[18].

While there is certainly the potential for impacts on other sectors, we believe these are likely small. Notably, due to data limitations, we have not analyzed the impact of the expansions on the American Samoa-based purse seine fleet, which previously fished within PRI. However, less than 5% of the fleet's harvest comes from this area, which suggests the impacts would be limited[46]. In addition, for the Hawaii longline tuna fishery, we only accounted for impacts on bigeye and yellowfin tuna catch rates. There may be negative impacts on catch rates for other species caught, such as mahi–mahi (*Coryphaena hippurus*) or sickle pomfret (*Taractichthys steindachneri*). We assume the relative impact of any such changes would be small because bigeye and yellowfin tuna account for about 80% of total revenue in the fishery. Although these fisheries constitute a small share of

activity within the monuments, their impacts are nonetheless important to consider.

Although we may under-estimate costs, we likely also under-estimate the potential benefits of the monuments. Previous ecological research suggests that large and remote protected areas may actually benefit the global fishing industry[3,14–17]. This argument is typically based on the idea that protected areas provide a sufficiently large refuge to recover and maintain mobile stocks, which can still be targeted when they swim beyond the borders of the protected area. In other words, the reserves provide a spillover of ecosystem services. Thus, an interesting question for future work is the degree to which species that are more resident in the monuments will benefit differentially over time. These changes could have important implications for small-scale, recreational, and cultural fishers in Hawaii. It may even be possible that there are spillover benefits for bigeye tuna because part of the PRI monument protects areas within known spawning grounds for this species[8–10]. It should also be noted that the two monument expansions were relatively recent (the PMNM expansion was only three years ago and we do not have observer data from 2018 or 2019): revenue or CPUE benefits might take time to develop.

Finally, our results may have important implications for other settings, especially parts of the world where large-scale marine protected areas have been implemented or proposed recently (including the ongoing United Nations conference on conservation of biological diversity in areas beyond national jurisdictions)[47]. For example, some of the debate over the impacts of large protected areas within the EEZs of the Parties to the Nauru Agreement countries[25] hinges on the degree to which fishing vessels will be able to easily relocate to other parts of the Pacific Ocean to catch a highly migratory species (skipjack tuna; *Katsuwonus pelamis*). Our results suggest that profitable fishing can still take place in areas outside the MPAs, provided these areas are geographically accessible.

## Methods

**Catch data**. Our measure of catch is the number of bigeye and yellowfin tuna, the primary target species responsible for over 77% of all revenue from Hawaii-based longline fishing in 2017. Estimates are based on Observer Program data, which reports the number of fish caught, not the number of fish kept. But for bigeye and

yellowfin tuna, these numbers are almost identical. In 2017, according to logbook summary reports, 98% of bigeye and yellowfin tuna caught were kept. Our measure of kilometers traveled is an approximation, since we only observe the location of fishing sets. We approximate distance traveled by calculating the distance covered by the sequence of sets on a trip plus the distances from both starting and ending sets to Honolulu (or Pago Pago).

**CPUE regressions**. In Table 1 we estimate regressions of the following form:

$$y_{i,t} = \beta_0 + \beta_1 \text{PRI}_t + \beta_2 \text{PMNM}_t + \mathbf{m}_t' \boldsymbol{\mu} + \mathbf{v}_i' \boldsymbol{\phi} + \mathbf{X}_{i,t}' \boldsymbol{\chi} + u_{i,t}, \quad (1)$$

where $y_{i,t}$ is the outcome variable of interest (such as catch per unit of effort) for vessel $i$ in time period $t$ (typically a day). $\beta_0$ is the standard intercept term and $\beta_1$ and $\beta_2$ are the main slope parameters of interest. $\text{PRI}_t$ is a dummy variable that takes the value of 0 for all dates prior to the PRI expansion and the value of 1 for all dates including and following the expansion date. The same logic applies to $\text{PMNM}_t$. $\mathbf{m}_t$ is a vector of month dummies, $\mathbf{v}_i$ is a vector of individual vessel dummies, and $\mathbf{X}_{i,t}$ is a vector of additional controls (oceanographic conditions, experimental fishing sets, etc.). Although we are indexing $\mathbf{X}_{i,t}$ with $i$ and $t$, not all of these variables will vary across vessels or across time. The specific controls we include are the following: an indicator dummy for whether the Western and Central Pacific region was closed due to the fleet reaching its TAC for that region (depending on the year this ranges from 0 to 31% of the days in the year), an indicator dummy for whether the Eastern region of the Pacific was closed to vessels over 24 m in length due to a binding TAC (75% of Hawaii-based vessels are less than 24 m and these closures ranged from 0 to 16% of the days in a year), an indicator variable for whether there was experimental research also being conducted as part of the fishing set, the monthly Niño 3.4 index provided by the National Center for Atmospheric Research[23], the same Niño 3.4 index lagged by one year, the Niño 3.4 index lagged by two years, and, finally, the Niño 3.4 index lagged by three years. The Niño 3.4 anomalies may be thought of as representing the average equatorial sea surface temperatures across the Pacific from about the dateline to the South American coast. The Niño 3.4 index typically uses a 5-month running mean, and El Niño or La Niña events are defined when the Niño 3.4 sea surface temperatures exceed ±0.4C for a period of 6 months or more. We estimate the slope coefficients using ordinary least squares estimation and we report heteroskedasticity-robust standard errors in all tables.

**Difference-in-differences regressions**. The main regression equation we estimate in Table 2 is the following:

$$y_{i,t} = \beta_0 + \beta_1 \text{MON}_t + \beta_2 \text{HI}_i + \beta_3 \text{HI}_i * \text{MON}_t + \mathbf{m}_t' \mu + \mathbf{a}_t' \psi + \mathbf{v}_i' \phi + \mathbf{X}_{i,t}' \chi + u_{i,t}, \quad (2)$$

where our primary outcome variable ($y_{i,t}$) is CPUE for vessel $i$ on day $t$. $\text{MON}_t$ is a dummy variable for the expansion of the monument (either PRI or PMNM, depending on the regression), $\text{HI}_i$ is a dummy variable indicating whether the vessel is part of the Hawaii-based longline tuna fleet, and $\text{HI}_i * \text{MON}_t$ is the two dummy variables interacted with (i.e., multiplied by) each other. $\mathbf{m}_t$ is a vector of month dummies, $\mathbf{a}_t$ is a vector of annual dummies, $\mathbf{v}_i$ is a vector of individual vessel dummies, and $\mathbf{X}_{i,t}$ is a vector of additional controls (oceanographic conditions, experimental fishing sets, etc.). $u_{i,t}$ is an unobserved error term. Our primary coefficient of interest is $\beta_3$, which can be interpreted as the causal effect of each monument expansion on the Hawaii tuna fleet, if our identifying assumptions hold true.

**Global Fishing Watch data**. GFW is an organization that provides access to information on commercial fishing activities, in particular information on the identity and location of fishing vessels. Many large vessels use a system known as the automatic identification system (AIS) to avoid collisions at sea, broadcast their location to port authorities and other vessels, and to view other vessels in their vicinity. AIS works through a very high frequency transceiver that automatically broadcasts vessel information such as current location and speed. This information is broadcast at regular intervals, in some cases as frequently as every 15 s. Vessels fitted with transceivers can be observed by AIS base stations and by satellites fitted with AIS receivers. The International Maritime Organization requires all large ships and all passenger ships to have an AIS transceiver on-board. The U.S. Coast Guard now requires it for all vessels larger than 65 ft. GFW obtains AIS data for fishing vessels and enables users with Internet access to monitor fishing activity globally, and to view individual vessel tracks. They also partner with academic researchers to provide more fine-scale data.

We requested individual vessel tracks for all vessels within the Hawaii longline fishery, based on publicly available permit data. We obtained GFW records for 148 different vessels but a number of these had yet to emit an AIS signal, leaving a total of 128 vessels with observed tracks. The dataset contains 5,592,202 observations of vessel locations from January 1st 2015 to December 31st 2017. Unfortunately, the GFW dataset is quite incomplete because vessels in the Hawaii longline fishery were not required to have AIS transceivers on-board until very recently. On March 1st 2016, the United States Coast Guard introduced a mandate requiring AIS on all U.S. vessels larger than 65 ft, which includes all of the pelagic longline vessels in Hawaii and some of the larger vessels in American Samoa. However, most vessels

appear to have initially ignored this mandate and did not obtain an AIS transceiver or switched it on until late 2016. We only have partial coverage of the fleet prior to the PMNM expansion (Supplementary Fig. 6): 71 vessels by August 2016 (approximately 50% of the active tuna and swordfish vessels that year) and 114 vessels by December 2017 (78.6% of active vessels for that year).

**Reporting summary**. Further information on research design is available in the Nature Research Reporting Summary linked to this article.

## Data availability
The aggregate logbook data that support the findings of this study are available from NOAA Fisheries https://www.fisheries.noaa.gov/resource/data/hawaii-longline-fishery-logbook-summary-reports.

The Observer Program data that support the findings of this study are available from NOAA Fisheries https://inport.nmfs.noaa.gov/inport/item/21854 but restrictions apply to the availability of these data, which contain business confidential information. Under the terms of a non-disclosure agreement with NOAA, J.L. cannot make these data publicly available.

The AIS vessel location data that support the findings of this study are available from Global Fishing Watch https://globalfishingwatch.org/. Under the terms of a data-sharing agreement with GFW, J.L. cannot make these data publicly available.

The source data underlying Figs. 2 and 3 are provided as a Source Data file. The source data underlying Figs. 1b and 4 are not publicly available according to the non-disclosure agreement with NOAA, described above. Summary data used to create Figs. 1b and 4 are provided in the Source Data file and the exact code used to create these summaries from the source data is also provided (see Code availability section).

## Code availability
All code used to generate the figures, tables, and results in this study are publicly available on GitHub at the following URL: https://github.com/lynham/monuments.

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

## Acknowledgements
We acknowledge NOAA's National Marine Fisheries Service for providing access to the observer data. Thanks to Aaron Bruner, Scott Edwards, Rachel Dacks, Eric Forney, Rhona Barr, David Kroodsma, Michel Chan, and Michael Tosatto for assistance, comments, and suggestions. All errors are our own. Lynham and Vilela would like to acknowledge Conservation Strategy Fund, who supported this article with a grant from the Pew Charitable Trusts and Pew Bertarelli Ocean Legacy.

## Author contributions
J.L. analyzed the observer data. J.L., A.N., and J.R. analyzed the GFW data. J.C.V.-D., T.V. and J.L. designed the map and figures. J.L., A.N., J.R., T.V., and J.C.V.-D. wrote the paper.

## Competing interests
Both J.L. and T.V. were indirectly funded by the Pew Charitable Trusts, which advocated in favor of the Papahānaumokuākea expansion. The funder did not play any role in the design, data collection, or analysis performed in this paper. The funder played a role in the conceptualization of the research question and reviewed an earlier copy of the paper. A.N., J.R., and J.C.V.-D. declare no competing interests.
