## [Peer Review File · Nature Communications]

Reviewers' comments:

Reviewer #1 (Remarks to the Author):

This paper estimates the economic impact of two of the largest protected. This is currently a topical as many such protected areas are being established around the world, we need a good understanding of their economic impacts to help guide policy and investment into probably managing them. This is an important contribution because, as far as I know, so far there are no deep analysis of this issue in the literature.

The authors applied a variety of indicators to help them come their conclusion based on strong evidence. A major conclusion is that “there have been minimal negative impacts from creating two of the largest protected areas on earth. This finding is backed up by conducting a number of counterfactual tests.

The contribution would be of importance to scientists in the specific field, which is very interdisciplinary in nature.

As far as I can tell you did not use profit as one of your economic indicators, why so? As the authors are well aware, for many economists, profit is the only thing that matters (I am not one of them). Given this, you need to explain why you did not use profit and what indicator you used i to capture the likely effects of cost of fishing on revenues (e.g., CPUE).

The authors could strengthen their argument by stressing more than they have done so far that the benefits of these large protected areas are not fully captured by catch and revenues. For instance, there is growing evidence that MPAs could mitigate the impact of climate change (e.g. Roberts et al. PNAS, 114 (24), 6167-6175; Cheung et al. Fish and Fisheries, 18(2), 254-263, doi:10.1111/faf.12177).

Reviewer #2 (Remarks to the Author):

This paper has the potential of attracting wide attention internationally, with implications for discussion of international policies. The science of course needs to be sound. Unfortunately, I do not see that the methods used here are appropriate to justify the claim made in the title, or allow us to make general statements about how fisheries may interact with the creation of large MPAs. To assess this properly, we would need to have more knowledge about the ecology of the MPAs. For

example, we would need to know about the productivity inside and outside the MPAs for target fish over time, and the role of the MPAs in the recruitment of fish to the fisheries. We would also need economic analyses that consider the overall economic health of the fisheries (not just analyses based on revenues). Moreover, there are insufficient data that would allow us to assess if the increased landings were due to oceanographic changes, the creation of the MPAs, or both. Another consideration is that the title implies that all potential economic impacts of the MPAs were considered, but this paper focussed only on the impact to the fisheries.

MPAs are an important tool for ecosystem based management. Assessing the economic impact of MPAs is one of the holy grail quests in marine science, especially as international targets for MPAs are currently 10%, with many researchers and governments justifiably recognising that these targets should be even more ambitious. MPAs do place constraints on human activities with potential effects (ranging from positive to negative) for a diversity of stakeholders. The various human activities (including also biodiversity protection and ecosystem based management) may not be compatible depending how the stake-holders value environmental, social and/or economic sustainability, and this can lead to intense conflicts inhibiting the creation of MPAs. Recent years have seen an increase in the number of very large Pacific MPAs and the Ross Ice Shelf MPA, etc, which together contributed much toward the steep increase in global area covered by MPAs, even if very few MPAs are strict marine protected areas. The locations of the two large MPAs that are the focus of this manuscript have been relatively less economic interest compared to other inshore locations within the USA EEZ, with fewer competing uses and stakeholders to be potentially affected by the MPAs.

What were the objectives of these MPAs, and have they been met over the past 10 years? The objectives were not focussed on the fisheries. The spillover of new recruits from within an MPA to fishing areas outside the MPA are usually considered as one of the ways that MPAs will benefit a fishery. What were the expectations about the impact of the MPAs on the productivity and recruitment of fish to the fisheries? Were there reasons to believe that recruitment would increase or decrease, or remain the same? This is important to consider, because the answers to these questions might be different for MPAs in other parts of the World, and may have implications for the degree of impact they might be expected to have on fisheries.

Use of fisheries revenues is a problematic way to estimate economic impact of the MPA on the economic health and sustainability of a fishery. Revenues do not convey information about expenses

or profits, subsidies, etc associated with fishing. They certainly do not assess the ecosystem impacts of fishing, nor do they have anything to say about how other harvestable resources are affected.

It takes time to see the effects of marine protected areas, and this will depend on many factors, e.g. the objectives of the MPA, the habitats in the MPA, the degree of habitat degradation, the life cycle of living resources being harvested (e.g. fish, seagrasses, etc), the time to recruitment into the

fishery, connectivity of the habitats in the system, the location of the source of recruitment, etc.

Where are the spawning and nursery habitats for the fish being harvested? Are they inside or outside of the MPAs? Without understanding any of these factors, we cannot understand why or why not the MPA has an economic impact on the fishery.

The title highlights that these are two of the largest MPAs in the World. The authors point out that the fished area lost by the MPA (less than 10% of the MPA area) is very small compared to the size of the MPAs. How large is the area that was used for fishing compared to fishing areas that were closed to fishing due to an MPA? If it is correct that there were no economic impacts of the formation of the MPA, do we know why? Is it because the MPA is so large compared to the small area that was used for fishing before the MPA was established (e.g. the large area that was not used as a fishing ground before is a source of new recruits to the population where fishing was actively taking place? Is it because the MPA is so large that we cannot detect it with the data we have?

In Line 247-248 The authors state that their results have implications for international policy because they show that “remote marine protected areas might be a very low-cost method of protecting biodiversity”. I think this is definitely overselling the value of the results of this paper.

Fishing is just one of a diversity of human activities that can impact biodiversity in remote marine areas. Deep sea mining, cable laying, and shipping are just few of the activities that can also occur in such remote areas, and all can threaten biodiversity, and could potentially produce greater revenues than fishing. Protecting marine areas from fishing, especially in remote areas, would also require effective monitoring control and surveillance (MCS), which may not be practicable in remote areas, especially where resources are limited (e.g. consider SIDS archipelagos, such as Tonga). Criticism also exists that the two MPAs that are the subject of this paper are considered paper parks. Other parts of the World may require more MCS than MPAs off the Hawaiian Islands.

Specific comments:

Line 1: I find that the title over-glorifies the results, and could be misunderstood by readers who do not understand ecosystem ecology, connectivity, and fisheries management. This MPA was designated to fill a variety of objectives (fisheries is not necessarily one of them). It is unknown how much of this large MPA may be needed to support a productive fishery. The authors explain that only a small part of these MPAs (4-9%) was actively fished before they were designated as MPAs. This means a very small area of fishing grounds became unavailable for fishing due to the creation of the MPA. A better title might be "Revenues from fishing fleets unchanged after 10 years of fishery closures in two Hawaiian MPAs".

Line 101: Is the El Niño index sufficient to estimate something as complex as the environmental suitability of tuna? Tuna are notoriously difficult to model, and changes in prey distribution can explain a lot. It is hard to say that by including an El Niño parameter in a regression can capture all of this.

Line 107: over-use of the continuous tense: "We also fail to reject the hypothesis that the distance the fleet is traveling is unchanged 108 following either expansion (SOM)".

Lines 116- 118 are unclear. Requires a few words of elaboration on excludability versus non interference

An additional concern is the assumption of "no interference": that there is no spillover from directly impacted to nonimpacted vessels.

Lines 109-201 read as if they were written in a great rush compared to the other sections. These deserve careful editing. Subject and verb do not agree in a few sentences explaining the regression methods. The methods explaining the regressions also seem a bit more detailed than other papers I have seen in Nature Communications. Some of this could be moved into the Supplementary Materials.

209-230 The idea to check for the change in the distance travelled before and after the expansion of

the MPA is an elegant one.

Line 207 For some reason, of all the data cited in this paper, only Global Fishing Watch data are referred to as "imperfect". GFW data are opportunistically harvested and processed from the global AIS data that countries make available. This of course does naturally limit how they can be used. Perhaps a more objective way to describe the Global Fishing Watch data is to include a neutral, short description of how the AIS data are acquired (made available by participating countries) and the natural limitations of the data for estimating fishing activity.

Reviewer #3 (Remarks to the Author):

This is an interesting study exploring the potential economic impact of the expansion of two MPAs to the fishing industry. The authors use different metrics and analyses to demonstrate that the expansion of the MPAs did not have a negative impact to economy of local fishery. I have two key suggestions and few rather minor comments which could be taken into account to further improve clarity of the ms.

My first concern is how to link the outputs with the actual reasoning behind the establishment of the MPA – conservation and protection-this part is largely ignored.

As an additional general comment, I would like to see if could somehow validate the estimates produced by the different methods applied here. Outputs could vary based on methods used e.g.

line107-108: "We also fail to reject the hypothesis that the distance the fleet is traveling is unchanged following either expansion (SOM)"

and lines 224-226: "we fail to reject the null hypothesis that distance traveled has remained constant since the monument expanded"

with such contradicting outputs posing some questions on whether any of datasets used might not offer a good quality of information.

One way to validate some of outputs would be by transferring the outputs into metric tones so that to have a standard basis for evaluation; e.g. based on the analysis the distance traveled per day increased by 12km while all CPEU metrics increased; so what that would mean in terms of catch? would be a realistic estimate based on the total distance traveled by the fleet?

other comments:

The title of the ms is misleading. The ms deals with economic impacts upon fishing industry but does not count for other sectors (e.g. tourism, aquaculture, etc).

The abstract seems very long. Should be reduced.

Lines 37-40 are exactly the same as written in the report <http://www.wpcouncil.org/wp-content/uploads/2016/06/2.13-wprmc-letter-to-president-obama.pdf> .I would be very careful on just copying a pasting full sentences even if a citation is used.

lines 56-57: so how is this translated in terms of conservation? e.g. expansion of no take zone resulted to higher efforts and high catches elsewhere (obviously in other productive areas so that to have no reduction of CPUE)? but higher efforts could be linked to higher by-catch rates (not refereeing to the small fleets such as the swordfish one!). What about the stocks inside the two MPAs? do you have evidence that they are increasing? The economic metrics by themselves could not be useful when talking about protection/conservation if there is no measure of the benefits to local biodiversity.

line71: this is by itself a very interesting finding. We need to understand whether the reference to the 10 millions of annual loss is linked to the actual effort that used to take place inside the MPAs. Might be needed to provide some values on metric tones to have a clear picture if the original claim of the loss is valid or not. If the Hawaiian fisheries generate annually 110-120 millions then how likely is that one eight of that is achieved in a very limited area? And if this is the case, then It seems rather paradox that all CPUE metrics seems to be suggest that less effort yields higher catch or the same effort now yields more catch. Could therefore expand the discussion on line232.

line 87: still there is about 25% which is not captured by your analysis and could alter the patterns.

line 227: You need to discuss the threshold selected.

Reviewer #1 (Remarks to the Author):

General comments

This paper estimates the economic impact of two of the largest protected. This is currently a topical as many such protected areas are being established around the world, we need a good understanding of their economic impacts to help guide policy and investment into probably managing them. This is an important contribution because, as far as I know, so far there are no deep analysis of this issue in the literature.

The authors applied a variety of indicators to help them come their conclusion based on strong evidence. A major conclusion is that “there have been minimal negative impacts from creating two of the largest protected areas on earth. This finding is backed up by conducting a number of counterfactual tests.

The contribution would be of importance to scientists in the specific field, which is very interdisciplinary in nature.

Specific comments

As far as I can tell you did not use profit as one of your economic indicators, why so? As the authors are well aware, for many economists, profit is the only thing that matters (I am not one of them). Given this, you need to explain why you did not use profit and what indicator you used i to capture the likely effects of cost of fishing on revenues (e.g., CPUE).

We agree that profits would be a better indicator of the direct impact of the expansion. However, fishing operations do not often disclose their cost structures (ice, bait, fuel, and labor, for example) nor profits. Yet, we believe that our approach provides a comprehensive analysis of the profitability of the fishery. For example, logbook data suggest that catch and revenues increased post-expansion. With increased revenues, the only way for profits to decrease is if costs had increased. The most likely way in which the expansion of the monuments could increase costs is by increasing distance traveled, (that is, increased fuel costs). Our analysis of observer data (CPUE and fishing locations) and GFW data (fishing locations and vessel tracks) do not suggest an increase in costs.

We agree, however, that this was not evident in our earlier version of the manuscript. In turn, we now explicitly state why we don't use profit data (**Lines 119-127**), and we have modified sections to better explain the thought process described above.

The authors could strengthen their argument by stressing more than they have done so far that the benefits of these large protected areas are not fully captured by catch and revenues. For instance, there is growing evidence that MPAs could mitigate the impact of climate change (e.g. Roberts et al. PNAS, 114 (24), 6167-6175; Cheung et al. Fish and Fisheries, 18(2), 254-263, doi:10.1111/faf.12177).

We completely agree with this point. Specifically, we have added two paragraphs to the beginning of the introduction (**Lines 25-51**) where we explain the diverse set of benefits that come with Marine Protected Areas and added the suggested references. We have also stated our motivation by explaining that these benefits are often “dispersed” among all members of society, but that the opportunity costs of MPAs are often concentrated among smaller groups, such as fishers, who tend to have high leverage in the decision-making process in marine conservation.

Reviewer #2 (Remarks to the Author)

We have carefully read your comments and suggestions and appreciate the objective skepticism with which our work was analyzed. One of the real benefits of the peer review process is getting a fresh pair of eyes to read through the entire manuscript. We believe all your points are valid and are happy to incorporate your suggestions into our work.

General comments

This paper has the potential of attracting wide attention internationally, with implications for discussion of international policies. The science of course needs to be sound. Unfortunately, I do not see that the methods used here are appropriate to justify the claim made in the title, or allow us to make general statements about how fisheries may interact with the creation of large MPAs. To assess this properly, we would need to have more knowledge about the ecology of the MPAs. For example, we would need to know about the productivity inside and outside the MPAs for target fish over time, and the role of the MPAs in the recruitment of fish to the fisheries. We would also need economic analyses that consider the overall economic health of the fisheries (not just analyses based on revenues). Moreover, there are insufficient data that would allow us to assess if the increased landings were due to oceanographic changes, the creation of the MPAs, or both. Another consideration is that the title implies that all potential economic impacts of the MPAs were considered, but this paper focussed only on the impact to the fisheries.

MPAs are an important tool for ecosystem based management. Assessing the economic impact of MPAs is one of the holy grail quests in marine science, especially as international targets for MPAs are currently 10%, with many researchers and governments justifiably recognising that these targets should be even more ambitious. MPAs do place constraints on human activities with potential effects (ranging from positive to negative) for a diversity of stakeholders. The various human activities (including also biodiversity protection and ecosystem based management) may not be compatible depending how the stake-holders value environmental, social and/or economic sustainability, and this can lead to intense conflicts inhibiting the creation of MPAs. Recent years have seen an increase in the number of very large Pacific MPAs and the Ross Ice Shelf MPA, etc, which together contributed much toward the steep increase in global area covered by MPAs, even if very few MPAs are strict marine protected areas. The locations of the two large MPAs that are the focus of this manuscript have been relatively less economic interest compared to other inshore locations within the USA EEZ, with fewer competing uses and stakeholders to be potentially affected by the MPAs.

What were the objectives of these MPAs, and have they been met over the past 10 years? The objectives were not focussed on the fisheries. The spillover of new recruits from within an MPA to fishing areas outside the MPA are usually considered as one of the ways that MPAs will benefit a fishery. What were the expectations about the impact of the MPAs on the productivity and recruitment of fish to the fisheries? Were there reasons to believe that recruitment would increase or decrease, or remain the same? This is important to consider, because the answers to these questions might be different for MPAs in other parts of the World, and may have implications for the degree of impact they might be expected to have on fisheries.

Use of fisheries revenues is a problematic way to estimate economic impact of the MPA on the economic health and sustainability of a fishery. Revenues do not convey information about expenses or profits, subsidies, etc associated with fishing. They certainly do not assess the ecosystem impacts of fishing, nor do they have anything to say about how other harvestable resources are affected. It takes time to see the effects of marine protected areas, and this will depend on many factors, e.g. the objectives of the MPA, the habitats in the MPA, the degree of habitat degradation, the life cycle of living resources being harvested (e.g. fish, seagrasses, etc), the time to recruitment into the fishery, connectivity of the habitats in the system, the location of the source of recruitment, etc. Where are the spawning and nursery habitats for the fish being harvested? Are they inside or outside of the MPAs? Without understanding any of these factors, we cannot understand why or why not the MPA has an economic impact on the fishery.

Thank you for these comments. We have tried to address them as follows. We have changed the title so that the focus is clearly only on economic impacts and only on the fishing industry. In terms of the original objectives of the MPAs, they have evolved over the years since the MPAs were created through four separate presidential declarations (2006, 2009, 2014, 2016). Nevertheless, the overarching purpose has been to protect “coral, fish, birds, marine mammals, and other flora and fauna including the endangered Hawaiian monk seal, the threatened green sea turtle, and the endangered leatherback and hawksbill sea turtles”.¹ All of the proclamations mention that “it is in the public interest to preserve the marine environment”.²

Since the purpose was primarily the protection of resources within the boundaries of the MPA, it was feared that these closures would have disproportionately negative impacts on the commercial fishing industry that used to fish there.³ Establishing whether that has indeed been the case is the primary objective of our research. Our focus is not on whether the MPAs have achieved their purpose of protecting biodiversity within the boundaries of the MPAs. We leave that question to others more qualified than ourselves. Our focus is solely on estimating the economic impact to the fishing fleet. Since all of the proclamations were decided on very quickly, there were not strong expectations about the impacts on productivity and recruitment of fish to the longline fishery. The degree to which the monuments provide a refuge to highly pelagic species like bigeye and yellowfin tuna is still a matter of scientific debate, but there were not strong expectations that there would be a recruitment spillover and this was not mentioned as justification.⁴

¹ <https://georgewbush-whitehouse.archives.gov/news/releases/2006/06/20060615-18.html> Alternatively: “many endemic species including corals, fish, shellfish, marine mammals, seabirds, water birds, land birds, insects, and vegetation not found elsewhere.” <https://georgewbush-whitehouse.archives.gov/news/releases/2009/01/20090106-6.html>

² <https://obamawhitehouse.archives.gov/the-press-office/2014/09/25/presidential-proclamation-pacific-remote-islands-marine-national-monumen> and <https://obamawhitehouse.archives.gov/the-press-office/2016/08/26/presidential-proclamation-papahanaumokuakea-marine-national-monument>

³ For example: “Expansion of the PMNM would adversely impact the Hawaii longline fishery”. <http://www.wpcouncil.org/wp-content/uploads/2016/06/2.13-wprmc-letter-to-president-obama.pdf>

⁴ Documents prepared by groups in favor of the 2nd expansion mention spillover as a possible benefit but without strong scientific support for the claim: “These findings suggest that individual tuna from different species could spend their entire life history inside the borders of a marine reserve if the area is large enough. It has been shown that female fish that are older and of larger size produce a higher number and a higher quality of eggs. These tuna would grow large and produce exponentially more eggs than smaller, unprotected individuals swimming outside the area of protection. Spillover effects of the fish that do swim outside of the area of protection would benefit fishermen.”

We completely agree with you that looking at revenue as the outcome variable can be misleading. Perhaps the previous version of the manuscript gave the impression that our focus is on revenue but that is definitely not the case. We agree with you that the focus should be on profitability and the best way that we can get at that, given the data that is available to us, is to use various measures of catch per unit effort. If the MPAs are affecting profitability by forcing the fleet to spend more money to catch the same amount of fish as before (or less) than this should be reflected through a change in CPUE. We have edited the manuscript to make it clearer that our focus is not on revenue but on profits: for example, “Aggregate increases in total catch and total revenue may be masking negative impacts from the monuments, such as forcing vessels to fish in less productive areas, to travel further, or to compete with more vessels in less space, all of which would increase the cost of fishing. Although we do not have data on profits in the industry because we do not have access to either individual or aggregated cost data, we can examine changes in CPUE. CPUE, measured as the ratio of catch to effort, is a proxy for fishery profitability because it relates the costs of fishing to the benefits of fishing. As an example, if a vessel has to expend twice as much effort to catch the same amount of fish, then CPUE would decline by 50%.” (Lines 119-127)

In terms of your question relating to spawning grounds, the Pacific Remote Islands MPA protects areas that are within the known spawning grounds for bigeye tuna. PMNM does not. We have now added some text to the manuscript to make this clear (Lines 48-49 and 313-315).

The title highlights that these are two of the largest MPAs in the World. The authors point out that the fished area lost by the MPA (less than 10% of the MPA area) is very small compared to the size of the MPAs. How large is the area that was used for fishing compared to fishing areas that were closed to fishing due to an MPA? If it is correct that there were no economic impacts of the formation of the MPA, do we know why? Is it because the MPA is so large compared to the small area that was used for fishing before the MPA was established (e.g. the large area that was not used as a fishing ground before is a source of new recruits to the population where fishing was actively taking place? Is it because the MPA is so large that we cannot detect it with the data we have?

This is one of the places where our text might have not been clear, we apologize. What we intended to say is that less than 10% of fishing sets took place inside the MPAs. Likewise, less than 10% of total catch came from inside the MPAs. In a sense, the industry can use these statistics to claim that they have lost 10% of their “fishing grounds”. But what we show is that their total catch has not declined by 10% once the MPAs are closed and they are not having to exert more effort than before. We believe this is because the target species are highly pelagic and over 90% of historical fishing grounds are still open to fishing. We have edited the text to make this “less than 10%” statistic clearer (Lines 102-106).

In Line 247-248 The authors state that their results have implications for international policy because they show that “remote marine protected areas might be a very low-cost method of protecting biodiversity”. I think this is definitely overselling the value of the results of this

https://www.researchgate.net/publication/305654430_Pu'uhonua_A_Place_of_Sanctuary_The_Cultural_and_Biological_Significance_of_the_proposed_expansion_for_the_Papahānaumokuākea_Marine_National_Monument/link/57980ad808ae33e89faede88/download

paper. Fishing is just one of a diversity of human activities that can impact biodiversity in remote marine areas. Deep sea mining, cable laying, and shipping are just few of the activities that can also occur in such remote areas, and all can threaten biodiversity, and could potentially produce greater revenues than fishing. Protecting marine areas from fishing, especially in remote areas, would also require effective monitoring control and surveillance (MCS), which may not be practicable in remote areas, especially where resources are limited (e.g. consider SIDS archipelagos, such as Tonga). Criticism also exists that the two MPAs that are the subject of this paper are considered paper parks. Other parts of the World may require more MCS than MPAs off the Hawaiian Islands.

Thank you for pointing this out. We were probably overselling our contribution. We have edited the title of the paper and the Discussion section to be more focused on what we actually demonstrate empirically and we have tried to remove any over-the-top generalizations.

Specific comments:

Line 1: I find that the title over-glorifies the results, and could be misunderstood by readers who do not understand ecosystem ecology, connectivity, and fisheries management. This MPA was designated to fill a variety of objectives (fisheries is not necessarily one of them). It is unknown how much of this large MPA may be needed to support a productive fishery. The authors explain that only a small part of these MPAs (4-9%) was actively fished before they were designated as MPAs. This means a very small area of fishing grounds became unavailable for fishing due to the creation of the MPA. A better title might be “Revenues from fishing fleets unchanged after 10 years of fishery closures in two Hawaiian MPAs”.

We appreciate the thoughtful comments about our title. We agree that it needed to be revisited and have modified it to better represent our results and conclusions.

Line 101: Is the El Niño index sufficient to estimate something as complex as the environmental suitability of tuna? Tuna are notoriously difficult to model, and changes in prey distribution can explain a lot. It is hard to say that by including an El Niño parameter in a regression can capture all of this.

ENSO events are known to drive the spatial distribution of tuna by causing a longitudinal shift in tuna's preferred thermal envelope (Lehodey et al., 1997, Aqorau et al., 2018). ENSO indices summarize fluctuations of sea surface temperature into an index that accurately represents ocean dynamics, especially in the Pacific Ocean. We use the NINO index not to account for the complex dynamics behind tuna biology, but to disentangle the effects of the expansions from environmental fluctuations that may occur at the same time as MPA creation. We are testing whether CPUE is higher or lower following the expansions, holding the ENSO index fixed. In other words controlling for any correlation between CPUE and the ENSO index. We agree that tuna are extremely difficult to model and we are not trying to do that.

Line 107: over-use of the continuous tense: “We also fail to reject the hypothesis that the distance the fleet is traveling is unchanged 108 following either expansion (SOM)”.

We have moved discussion of this null finding to the SOM as it distracts from our main message.

Lines 116- 118 are unclear. Requires a few words of elaboration on excludability versus non interference. An additional concern is the assumption of “no interference”: that there is no spillover from directly impacted to nonimpacted vessels.

We have completely overhauled this section of the paper. We now try to explain the two assumptions as simply and as clearly as possible. We give examples of how the assumptions could be violated and honestly assess which assumptions are most and least likely to hold in our different regression models (**Lines 147-239**).

Lines 109-201 read as if they were written in a great rush compared to the other sections. These deserve careful editing. Subject and verb do not agree in a few sentences explaining the regression methods. The methods explaining the regressions also seem a bit more detailed than other papers I have seen in Nature Communications. Some of this could be moved into the Supplementary Materials.

We have made substantial revisions to this part of the article. Hopefully it reads a lot better now. We have moved the detailed discussion of the regressions to the new Methods section and to the Supplementary Materials.

209-230 The idea to check for the change in the distance travelled before and after the expansion of the MPA is an elegant one.

Thank you.

Line 207 For some reason, of all the data cited in this paper, only Global Fishing Watch data are referred to as “imperfect”. GFW data are opportunistically harvested and processed from the global AIS data that countries make available. This of course does naturally limit how they can be used. Perhaps a more objective way to describe the Global Fishing Watch data is to include a neutral, short description of how the AIS data are acquired (made available by participating countries) and the natural limitations of the data for estimating fishing activity.

We appreciate this consideration. We have removed the subjective “imperfect” qualifier and expanded on the way AIS data is derived in the Methods section (**Lines 363-377**). Additionally, we explain why only a portion of the vessels present in our observer dataset could be matched to AIS data from GFW (**Lines 378-388**).

References

Aqorau, T., Bell, J. & Kittinger, J. N. Good governance for migratory species. *Science* 389 361, 1208.2–1209 (2018).

Lehodey, P., Bertignac, M., Hampton, J., Lewis, A. & Picaut, J. El Niño southern 386 oscillation and tuna in the western pacific. *Nature* 389, 715–718 (1997).

Reviewer #3 (Remarks to the Author):

This is an interesting study exploring the potential economic impact of the expansion of two MPAs to the fishing industry. The authors use different metrics and analyses to demonstrate that the expansion of the MPAs did not have a negative impact to economy of local fishery. I have two key suggestions and few rather minor comments which could be taken into account to further improve clarity of the ms.

My first concern is how to link the outputs with the actual reasoning behind the establishment of the MPA – conservation and protection-this part is largely ignored.

Thank you for this comment. We have added some text (**Lines 46-51**) to better explain the original reasoning behind the creation of the MPAs.

As an additional general comment, I would like to see if could somehow validate the estimates produced by the different methods applied here. Outputs could vary based on methods used e.g. line107-108: "We also fail to reject the hypothesis that the distance the fleet is traveling is unchanged following either expansion (SOM)" and lines 224-226: "we fail to reject the null hypothesis that distance traveled has remained constant since the monument expanded" with such contradicting outputs posing some questions on whether any of datasets used might not offer a good quality of information.

We apologize for the confusion. Another reviewer also drew our attention to how these statements appear to contradict each other, even though they do not. We have moved the discussion of these null findings to the SOM since they distract from the main message of the paper. All of the methods and data-sets used provide the same overall message: the MPA expansions have not caused CPUE to decrease.

One way to validate some of outputs would be by transferring the outputs into metric tones so that to have a standard basis for evaluation; e.g. based on the analysis the distance traveled per day increased by 12km while all CPEU metrics increased; so what that would mean in terms of catch? would be a realistic estimate based on the total distance traveled by the fleet?

This is a great idea and something we would love to do. Unfortunately, the data provided to us by NOAA records the number of fish caught and not the weight (tonnes). We have repeatedly asked for weight data which is recorded in confidential logbooks but, so far, NOAA has not been willing to share it with us.

Specific comments

The title of the ms is misleading. The ms deals with economic impacts upon fishing industry but does not count for other sectors (e.g. tourism, aquaculture, etc).

Very good point. The title has been modified such that it specifies that we are investigating impacts on just the longline fishing industry.

The abstract seems very long. Should be reduced.

Thank you for pointing this out. We did not have enough time to reformat the abstract/intro paragraph when the article was transferred from Nature to Nature Communications. The abstract has now been modified and shortened for improved readability. It is now 131 words.

Lines 37-40 are exactly the same as written in the report <http://www.wpcouncil.org/wp-content/uploads/2016/06/2.13-wprmc-letter-to-president-obama.pdf>. I would be very careful on just copying a pasting full sentences even if a citation is used.

We appreciate the detailed review of our text and the original sources. Our intention in including this text verbatim was to present the exact words in the letter and try to remain impartial. We understand that this approach of exactly quoting another text is not common in the scientific literature and we now paraphrase the sentences. However, we use quotation marks to explicitly state words that were taken verbatim from the letter to provide transparency on the source of these values.

lines 56-57: so how is this translated in terms of conservation? e.g. expansion of no take zone resulted to higher efforts and high catches elsewhere (obviously in other productive areas so that to have no reduction of CPUE)? but higher efforts could be linked to higher by-catch rates (not refereeing to the small fleets such as the swordfish one!). What about the stocks inside the two MPAs? do you have evidence that they are increasing? The economic metrics by themselves could not be useful when talking about protection/conservation if there is no measure of the benefits to local biodiversity.

First, we agree that the implications for conservation are of utmost importance and have added a mention of this in our Introduction (**Lines 25-38**) and in the Discussion section (**Lines 305-315**). NOAA did not provide us with data on turtles and marine mammals so we can not evaluate whether the expansions caused bycatch rates to increase or decrease. Likewise, as now stated in our introduction, these MPAs were not intended to provide fisheries benefits. In fact, the objective of the PMNM is “To forever protect and perpetuate ecosystem health and diversity and Native Hawaiian cultural significance of Papahānaumokuākea.” (PNMN Website).

Second, previous work on similar systems (spatial closures for tuna fisheries in the Pacific) suggest that spatial closures are unlikely to have an effect on tuna stocks (Sibert et al., 2012). However, we must recognize that recent evidence suggests that large-scale MPAs such as the Phoenix Islands Protected Area may provide some protection to spawning tuna (Hernández et al., 2019). Whether this translates to increases in stocks is still uncertain, but is highly unlikely due to the fishing-the line effect observed in most pelagic MPAs, which would allow vessels to harvest spawning fish as they enter or leave these areas. This point, however, is worth mentioning and we have included it in our introduction (**Lines 48-49**) to provide a clear understanding of the benefits and costs of the closure.

line71: this is by itself a very interesting finding. We need to understand whether the reference to the 10 millions of annual loss is linked to the actual effort that used to take place inside the MPAs. Might be needed to provide some values on metric tones to have a clear picture if the original claim of the loss is valid or not. If the Hawaiian fisheries generate annually 110-120 millions then how likely is that one eight of that is achieved in a very

limited area? And if this is the case, then It seems rather paradox that all CPUE metrics seems to be suggest that less effort yields higher catch or the same effort now yields more catch. Could therefore expand the discussion on line232.

We agree that this is an important point. We now explain how the “\$10 million” estimate was calculated but it was basically as you suspect: 10% of catch was coming from this area so 10% of catch will be lost, which is equivalent to \$10 million.

We were surprised to observe an increase in CPUE even after our counterfactual analysis. This is why we try to be careful in arguing that our results certainly reject the claim that the industry is worse off but it would be premature to argue that they are better off as a result of the expansions. As suggested, we have expanded the discussion that used to start on line 232 and now starts on **Line 283**. We disagree slightly with the statement that the PMNM MPA is a very limited area. This MPA is larger than the landmass of Peru or the state of Alaska.

line 87: still there is about 25% which is not captured by your analysis and could alter the patterns.

Correct. We now mention this as a potential caveat in the Discussion section (**Lines 298-304**).

line 227: You need to discuss the threshold selected.

This specific hypothesis test was confusing a lot of readers. We have deleted it.

References

Hernandez, C. M. et al. Evidence and patterns of tuna spawning inside a large no-take marine protected area. *Scientific reports* 9, 10772 (2019).

Sibert, J., Senina, I., Lehodey, P. & Hampton, J. Shifting from marine reserves to maritime zoning for conservation of pacific bigeye tuna (*thunnus obesus*). *Proceedings of the National Academy of Sciences* 109, 18221–18225 (2012).

Reviewers' comments:

Reviewer #1 (Remarks to the Author):

The authors have adequately addressed my comments and therefore I've recommended that the paper be published in Nature Communications.

Reviewer #2 (Remarks to the Author):

I thank the authors for their revision.

I find that the authors addressed most of my suggestions in the revision. I do have a few remaining important critiques.

I appreciate that the authors revised the title, but I unfortunately still think that the title still seems to over-glorify the results. The paper does not examine the impact on the entire long line fishing industry. It only examines impacts of those operating in the region of the PMNM and PRI. It states economic impact in the title, but this is not general economic impact, the paper only looks at revenues and CPUE. Although less attention grabbing, I would change the title to something humbler such as some succinct version of "Assessing the impacts of the World's two largest MPAs on their regional long line fleet fishery revenues and catch per unit effort".

I also found a number of places in the text where the authors made conclusions that could not be justified because they extrapolated beyond the extent of the analyses and the context of their study. Below I provide examples of those from the manuscript, along with other detailed comments.

On another note, I found it very difficult to review this revised manuscript because the authors did not highlight line numbers where I could find the changes they made in the text. I needed to hunt for the changes. Although I expect that this saved the authors time, I am sure this created some extra work for the reviewers.

Specific comments:

Line 30-31

"MPAs also impose large opportunity costs by displacing existing activities, notably fishing effort, or preventing the development of new uses, such as deep sea mining."

Note: the lost opportunity cost will likely be greater if the fish stock collapses.

I therefore find this quote misleading because it fails to acknowledge that the MPAs (at least the well designed and well managed ones) have been shown to create more opportunities and benefits because of the increased resilience and productivity of the fisheries and ecosystems in the long run. I think it would suffice to say that MPAs restrict the use of some human activities in an area in order to bolster the benefits of nature to society.

Line 32

“The benefits derived from MPAs are often dispersed across many different stake33 holders and take time to develop, while the costs are often incurred immediately by specific groups, such as fishers.”

I suggest rephrasing this to

“The benefits derived from MPAs are often dispersed across fishers and a diversity of other stakeholders, and take time to develop. The closures however will require a change in the activities of some stakeholders, such as fishers, and this may come at a nearer term revenue loss to some.”

Line 41:

“...–the third- and fifth-largest protected areas in the world, respectively–offer the unique opportunity to rigorously examine the economic costs of large MPAs to the fishing industry.”

I am surprised that this paper uses such a glass is half empty approach to discussing MPA and economic impacts. One could easily search and replace this the word economic cost with “economic benefit”.

Here and wherever else possible in this MS, I would exchange the word “cost” with “change in revenue and effort” or similar. This is a neutral way to refer to the volume of water in the glass.

Lines 277-278:

“We find no observable declines in catch and, in fact, both aggregate CPUE and revenue increase 278 following each expansion (although these gains attenuate in more careful counterfactual analyses).”

What is the evidence that the gains attenuated in more careful counterfactual analyses? Were there “less careful” analyses elsewhere? I am afraid I do not understand these lines.

I think the authors need to address the age of the reserves and the role that might play in the revenue benefits to the fishery over time

Lines 280-281:

“These results contradict earlier predictions of large economic losses and show that, 280 under certain conditions, carefully-crafted MPAs can protect biodiversity and ecosystems 281 without compromising the profitability of fisheries. “

Unfortunately I do not think your results can be used to show that there are ecosystem benefits from these MPAs, nor tested for how carefully they were “crafted”.

I suggest a humbler conclusion that refers to what you tested, such as:

These results contradict earlier predictions of large economic losses and show that, these two MPAs, though developed with the objective to protect rare iconic species, did not diminish the revenues or catch per unit effort of fisheries.”

Lines 316-318

“Finally, our results have important implications for other settings, especially parts of the 317 world where large-scale marine protected areas have been implemented or proposed recently 318 (including the ongoing United Nations conference on conservation of biological diversity in areas beyond national jurisdictions)... Our results suggest that, assuming access to the high seas is relatively unencumbered, fishing 324 captains are highly adept at relocating and finding other productive waters.”

Likewise, I suggest a humbler conclusion that reflects what was tested in the paper.

I suggest

“Finally, our results MAY have important implications for other settings, especially parts of the world where large-scale marine protected areas have been implemented or proposed recently (including the ongoing United Nations conference on conservation of biological diversity in areas beyond national jurisdictions)... Our results suggest that fishing revenues could still increase in areas outside the MPAs.”

Figure 1 B could be simplified, as the bars are mostly filled with gray.

Figure 2, Y Axis, why not 100,000 instead of e.g. 1e+05, which is only one more digit and a simpler way to present numbers

Throughout the manuscript, I would like to suggest replacing the word “counterfactual” with “control”. This is probably more intuitive for most readers. It also aligns with your connections to the application of the BACI approach (BACI for Before After Control Impact, not Before After Counterfactual Impact).

Reviewer #3 (Remarks to the Author):

The authors have done a great effort to deal with the comments raised. I acknowledge that due to data limitation some issues could not be addressed in a different way; Still I appreciate that limitations, gaps or challenges are now clearly mentioned. I have only few minor suggestion in case the authors would like to consider:

1. Lines 109-110. I'm not sure if it because of my English but I think that this sentence might be a bit strict. I would rephrase it. I think you should "smooth" a bit the second part of the sentence: a limited number of trips have been actually affected? Or could support your suggestion by using the annual range of 4-9% so that in some years 5% reduction of trips is classic and thus would not accept to be something very hard to be done..
2. A recent paper from Europe: Mazaris et al., 2019 Science of The Total Environment 677, 418-426, demonstrates that a number of threats are still documents within most MPAs with fishery one of most frequent one. Your case study (large MPAs, great catches, enforcement and expansion) could be considered as an example towards exploring impacts of new decisions or even setting a standardized methodology for assessing patterns and changes.
3. 316-324: This is last paragraph of the ms, I love the first message but I;m not sure whether a statement like 323-324 provides any clear message for spatial planning or conservation...

Response to Reviewers' comments:

Reviewer #1 (Remarks to the Author):

The authors have adequately addressed my comments and therefore I've recommended that the paper be published in Nature Communications.

Response: Thank you.

Reviewer #2 (Remarks to the Author):

I thank the authors for their revision.

I find that the authors addressed most of my suggestions in the revision. I do have a few remaining important critiques.

I appreciate that the authors revised the title, but I unfortunately still think that the title still seems to over-glorify the results. The paper does not examine the impact on the entire long line fishing industry. It only examines impacts of those operating in the region of the PMNM and PRI. It states economic impact in the title, but this is not general economic impact, the paper only looks at revenues and CPUE. Although less attention grabbing, I would change the title to something humbler such as some succinct version of "Assessing the impacts of the World's two largest MPAs on their regional long line fleet fishery revenues and catch per unit effort".

Response: Thank you for this suggestion. We have changed the title to a modified version of what you suggest: "Impact of Two of the World's Largest Protected Areas on Catch-Per-Unit-Effort in the Hawaii Longline Fishery". This slightly exceeds Nature Communication's 15-word limit for titles.

I also found a number of places in the text where the authors made conclusions that could not be justified because they extrapolated beyond the extent of the analyses and the context of their study. Below I provide examples of those from the manuscript, along with other detailed comments.

On another note, I found it very difficult to review this revised manuscript because the authors did not highlight line numbers where I could find the changes they made in the text. I needed to hunt for the changes. Although I expect that this saved the authors time, I am sure this created some extra work for the reviewers.

Response: We sincerely apologize for this. We should have included the line numbers of changes to make it easier to identify them in the manuscript. We now include them for all of the changes in the latest version.

Specific comments:

Line 30-31

"MPAs also impose large opportunity costs by displacing existing activities, notably fishing effort, or preventing the development of new uses, such as deep sea mining."

Note: the lost opportunity cost will likely be greater if the fish stock collapses.

I therefore find this quote misleading because it fails to acknowledge that the MPAs (at least the well designed and well managed ones) have been shown to create more opportunities and benefits because of the increased resilience and productivity of the fisheries and ecosystems in the long run. I think it would suffice to say that MPAs restrict the use of some human activities in an area in order to bolster the benefits of nature to society.

Response: Good points. A previous referee complained that we were consistently overstating the potential benefits of MPAs so we have probably swung too much in the opposite direction now. We completely agree with you and have modified the sentence as follows: “MPAs might also impose short-run opportunity costs by displacing existing activities, notably fishing effort, or by preventing the development of new uses, such as deep-sea mining.” [Lines 31-33] This emphasizes that the costs are possible, not guaranteed, and more likely to be in the short-run only.

Line 32

“The benefits derived from MPAs are often dispersed across many different stake33 holders and take time to develop, while the costs are often incurred immediately by specific groups, such as fishers.”

I suggest rephrasing this to

“The benefits derived from MPAs are often dispersed across fishers and a diversity of other stakeholders, and take time to develop. The closures however will require a change in the activities of some stakeholders, such as fishers, and this may come at a nearer term revenue loss to some.”

Response: We have made the suggested change. [Lines 33-34]

Line 41:

“...–the third- and fifth-largest protected areas in the world, respectively–offer the unique opportunity to rigorously examine the economic costs of large MPAs to the fishing industry.” I am surprised that this paper uses such a glass is half empty approach to discussing MPA and economic impacts. One could easily search and replace this the word economic cost with “economic benefit”.

Here and wherever else possible in this MS, I would exchange the word “cost” with “change in revenue and effort” or similar. This is a neutral way to refer to the volume of water in the glass.

Response: Thank you for pointing this out. As we mention above, we have probably gone too “glass half empty” in response to earlier referee comments at a different journal. We have replaced “economic costs” with “economic effects” [Line 43] and done the same throughout the text wherever appropriate:

- “Numerous claims have been made about the costs of these protected areas to the fishing industry” → “Numerous claims have been made about the impacts of these protected areas on the fishing industry” [Lines 14-16]
- “could lead to large economic costs” → “could potentially lead to economic costs” [Lines 54-56]

Lines 277-278:

“We find no observable declines in catch and, in fact, both aggregate CPUE and revenue increase 278 following each expansion (although these gains attenuate in more careful counterfactual analyses).”

What is the evidence that the gains attenuated in more careful counterfactual analyses? Were there “less careful” analyses elsewhere? I am afraid I do not understand these lines.

Response: By “less careful” analysis we mean our own analysis that did not include control fisheries. Sorry for the confusion. The new sentences are now: “We find no observable declines in catch and, in fact, both aggregate CPUE and revenue increase following each expansion (although the increases are smaller and, in some cases, statistically indistinguishable from zero when we include our control fisheries).” **[Lines 278-281]**

I think the authors need to address the age of the reserves and the role that might play in the revenue benefits to the fishery over time

Response: Good point. We have added the following sentence: “It should also be noted that the two monument expansions were relatively recent (the PMNM expansion was only three years ago and we do not have observer data from 2018 or 2019): revenue or CPUE benefits might take time to develop.” **[Lines 318-321]**

Lines 280-281:

“These results contradict earlier predictions of large economic losses and show that, 280 under certain conditions, carefully-crafted MPAs can protect biodiversity and ecosystems 281 without compromising the profitability of fisheries.”

Unfortunately I do not think your results can be used to show that there are ecosystem benefits from these MPAs, nor tested for how carefully they were “crafted”. I suggest a humbler conclusion that refers to what you tested, such as: These results contradict earlier predictions of large economic losses and show that, these two MPAs, though developed with the objective to protect rare iconic species, did not diminish the revenues or catch per unit effort of fisheries.”

Response: We have changed the sentence as suggested, with some slight modifications. **[Lines 281-284]**

Lines 316-318

“Finally, our results have important implications for other settings, especially parts of the 317 world where large-scale marine protected areas have been implemented or proposed recently 318 (including the ongoing United Nations conference on conservation of biological diversity in areas beyond national jurisdictions)... Our results suggest that, assuming access to the high seas is relatively unencumbered, fishing 324 captains are highly adept at relocating and finding other productive waters.”

Likewise, I suggest a humbler conclusion that reflects what was tested in the paper. I suggest

“Finally, our results MAY have important implications for other settings, especially parts of the world where large-scale marine protected areas have been implemented or proposed recently (including the ongoing United Nations conference on conservation of biological diversity in areas beyond national jurisdictions)... Our results suggest that fishing revenues could still increase in areas outside the MPAs.”

Response: Done, with a slight modification: “Finally, our results may have important implications for other settings, especially parts of the world where large-scale marine protected areas have been implemented or proposed recently (including the ongoing United Nations conference on conservation of biological diversity in areas beyond national jurisdictions)... Our results suggest that profitable fishing can still take place in areas outside the MPAs.” [Lines 322-325] and [Lines 329-330]

Figure 1 B could be simplified, as the bars are mostly filled with gray.

Response: We are going to keep this figure as is. The large gray area in each column quickly conveys to the reader that over 90% of fishing was taking place in the gray area of the map (i.e. outside of the MPAs), even when the MPAs were open to fishing.

Figure 2, Y Axis, why not 100,000 instead of e.g. 1e+05, which is only one more digit and a simpler way to present numbers

Response: Done.

Throughout the manuscript, I would like to suggest replacing the word “counterfactual” with “control”. This is probably more intuitive for most readers. It also aligns with your connections to the application of the BACI approach (BACI for Before After Control Impact, not Before After Counterfactual Impact).

Response: Done. Examples below.

- This technique, described further below, uses set-level data from NOAA's Observer Program to construct a counterfactual for what would have happened in the affected fisheries in the absence of each expansion. → This technique, described further below, uses set-level data from NOAA's Observer Program to construct a control for what would have happened in the affected fisheries in the absence of each expansion. [Lines 79-81]
- We find that the expansions had statistically insignificant effects on catch rates, with the exception of a significant drop in catch-per-fishing-trip; the latter is caused by a smaller increase in trip length and total hooks deployed per trip in the affected fishery versus in the counterfactual. → We find that the expansions had statistically insignificant effects on catch rates, with the exception of a significant drop in catch-per-fishing-trip; the latter is caused by a smaller increase in trip length and total hooks deployed per trip in the affected fishery versus in the control fishery. [Lines 81-84]
- A criticism of the literature on marine reserve impacts is that the absence of an appropriate counterfactual makes it challenging to disentangle reserve impacts from unobserved factors that are changing at the same time. → A criticism of the literature on marine reserve impacts is that the absence of an appropriate control makes it challenging to disentangle reserve impacts from unobserved factors that are changing at the same time. [Lines 151-153]
- The no interference assumption states that when using a counterfactual approach, there can be no interference or spillover from the impacted fishery to the

counterfactual fishery. → The no interference assumption states that when using a control-impact approach, there can be no interference or spillover from the impacted fishery to the control fishery. [Lines 159-161]

- This requires that the treated group of vessels does not change the behavior of the counterfactual group of vessels following reserve implementation. A classic example of a violation of the no interference assumption would be if vessels impacted by the reserve displace the counterfactual vessels and force them to fish in less productive waters, causing their CPUE to decline. → This requires that the treated group of vessels does not change the behavior of the control group of vessels following reserve implementation. A classic example of a violation of the no interference assumption would be if vessels impacted by the reserve displace the control vessels and force them to fish in less productive waters, causing their CPUE to decline. [Lines 161-165]
- In order to make credible statements about the causal impacts of the expansions, we need to find a counterfactual for CPUE in the Hawaii tuna fishery. In other words, what would the trend in CPUE have been if the monuments had not been expanded? The counterfactual or control must be influenced by the same unobserved factors that might be correlated with the monument expansions, such as changes in oceanographic conditions or management rules, but remain unaffected by the expansions themselves (allowing us to control for these unobserved factors and satisfy the excludability assumption). → In order to make credible statements about the causal impacts of the expansions, we need to find a control for CPUE in the Hawaii tuna fishery. In other words, what would the trend in CPUE have been if the monuments had not been expanded? The control must be influenced by the same unobserved factors that might be correlated with the monument expansions, such as changes in oceanographic conditions or management rules, but remain unaffected by the expansions themselves (allowing us to control for these unobserved factors and satisfy the excludability assumption). [Lines 173-179]
- We capitalize on incidental catch of bigeye and yellowfin tuna in two closely related fisheries to construct our counterfactuals--the Hawaii longline swordfish (*Xiphias gladius*) fleet for the PRI expansion and the American Samoa longline albacore tuna (*Thunnus alalunga*) fleet for the PMNM expansion (the dashed grey lines in Figure 4 show the mean of the four CPUE measures for these counterfactual fisheries). → We capitalize on incidental catch of bigeye and yellowfin tuna in two closely related fisheries to construct our controls--the Hawaii longline swordfish (*Xiphias gladius*) fleet for the PRI expansion and the American Samoa longline albacore tuna (*Thunnus alalunga*) fleet for the PMNM expansion (the dashed grey lines in Figure 4 show the mean of the four CPUE measures for these control fisheries). [Lines 179-183]
- The Hawaii swordfish fleet is a good counterfactual for the PRI expansion for two reasons. → The Hawaii swordfish fleet is a good control for the PRI expansion for two reasons. [Line 184]
- When we analyze the PMNM expansion, we use incidental catch of bigeye and yellowfin tuna in the American Samoa albacore fishery as our counterfactual measure of CPUE. → When we analyze the PMNM expansion, we use incidental

catch of bigeye and yellowfin tuna in the American Samoa albacore fishery as our control measure of CPUE. **[Lines 202-203]**

- In Figure S6 in the SOM, we show that the catch rates of bigeye and yellowfin tuna are strongly positively correlated across the Hawaii tuna and American Samoa albacore tuna fisheries, confirming our claim that the American Samoa fleet is a plausible counterfactual. → In Figure S6 in the SOM, we show that the catch rates of bigeye and yellowfin tuna are strongly positively correlated across the Hawaii tuna and American Samoa albacore tuna fisheries, confirming our claim that the American Samoa fleet is a plausible control. **[Lines 213-216]**
- Both counterfactuals also likely satisfy the no interference assumption. → Both controls also likely satisfy the no interference assumption. **[Lines 223]**
- These robustness checks give us confidence that the no interference assumption holds, and that each of the surrogate fisheries are valid counterfactuals. → These robustness checks give us confidence that the no interference assumption holds, and that each of the surrogate fisheries are valid controls. **[Lines 234-236]**
- Further investigation reveals that this decrease is due to the length of fishing trips growing more slowly in the Hawaii tuna fleet, relative to the counterfactual, following the expansion (SOM). → Further investigation reveals that this decrease is due to the length of fishing trips growing more slowly in the Hawaii tuna fleet, relative to the control, following the expansion (SOM). **[Lines 258-260]**
- By combining three different data sources and exploiting novel counterfactuals, we provide a detailed account of how the creation of two of the world's largest MPAs affected the Hawaii-based longline fishing industry. → By combining three different data sources and exploiting novel controls, we provide a detailed account of how the creation of two of the world's largest MPAs affected the Hawaii-based longline fishing industry. **[Lines 275-277]**
- The horizontal dashed grey lines represent the same calculation (the mean of all four standardized catch-per-unit-effort measures) for two counterfactual fisheries: the segment following PRI expansion but prior to PMNM expansion is calculated using Hawaii swordfish fleet data and the post-PMNM segment is calculated using American Samoa tuna fleet data. → The horizontal dashed grey lines represent the same calculation (the mean of all four standardized catch-per-unit-effort measures) for two control fisheries: the segment following PRI expansion but prior to PMNM expansion is calculated using Hawaii swordfish fleet data and the post-PMNM segment is calculated using American Samoa tuna fleet data. **[Lines 562-566]**

Reviewer #3 (Remarks to the Author):

The authors have done a great effort to deal with the comments raised. I acknowledge that due to data limitation some issues could not be addressed in a different way; Still I appreciate that limitations, gaps or challenges are now clearly mentioned. I have only few minor suggestion in case the authors would like to consider:

1. Lines 109-110. I'm not sure if it because of my English but I think that this sentence might be a bit strict. I would rephrase it. I think you should "smooth" a bit the second part of the sentence: a limited number of trips have been actually affected? Or could support your suggestion by using

the annual range of 4-9% so that in some years 5% reduction of trips is classic and thus would not accept to be something very hard to be done..

Response: Thanks for pointing this out. We have changed the sentence from “Taken as a whole, these results suggest that any negative effects of the monument expansion on the fishery are likely relatively small, as the few affected trips can move elsewhere.” To “Taken as a whole, Figure 1 suggests that a very small fraction of total fishing effort has been displaced by the monument expansions.” **[Lines 112-113]**

2. A recent paper from Europe: Mazaris et al., 2019 Science of The Total Environment 677, 418-426, demonstrates that a number of threats are still documents within most MPAs with fishery one of most frequent one. Your case study (large MPAs, great catches, enforcement and expansion) could be considered as an example towards exploring impacts of new decisions or even setting a standardized methodology for assessing patterns and changes.

Response: Thank you for pointing this out and bringing this recent publication to our attention. We now cite it in the introduction. **[Lines 31-33]**

3. 316-324: This is last paragraph of the ms, I love the first message but I;m not sure whether a statement like 323-324 provides any clear message for spatial planning or conservation...

Response: The other reviewer also did not like this sentence. We have changed it so it now better reflects our findings and relevance to future marine spatial planning efforts. **[Lines 329-330]**